# Text-Adaptive Generative Adversarial Networks: Manipulating Images with Natural Language

**Seonghyeon Nam, Yunji Kim, and Seon Joo Kim**
Yonsei University
{shnnam,kim_yunji,seonjookim}@yonsei.ac.kr

## Abstract

This paper addresses the problem of manipulating images using natural language description. Our task aims to semantically modify visual attributes of an object in an image according to the text describing the new visual appearance. Although existing methods synthesize images having new attributes, they do not fully preserve text-irrelevant contents of the original image. In this paper, we propose the text-adaptive generative adversarial network (TAGAN) to generate semantically manipulated images while preserving text-irrelevant contents. The key to our method is the text-adaptive discriminator that creates word-level local discriminators according to input text to classify fine-grained attributes independently. With this discriminator, the generator learns to generate images where only regions that correspond to the given text are modified. Experimental results show that our method outperforms existing methods on CUB and Oxford-102 datasets, and our results were mostly preferred on a user study. Extensive analysis shows that our method is able to effectively disentangle visual attributes and produce pleasing outputs.

## 1 Introduction

Taking pictures has become a big part of people's life ever since the emergence of smartphones as the everyday device. With this trend, the demand for manipulating or editing images is growing, to make photos to look better or to meet a user's need. While one can use commercially available tools such as the Photoshop to manipulate images, it is difficult for a non-expert user to use those tools for image manipulation. Additionally, manipulating images on a mobile device would be difficult as the interface on mobile devices is rather limited.

Automatic image manipulation can remedy this difficulty by automatically changing various aspects of images from low-level color/texture [1, 2] to high-level semantics [3] without tedious human operation. With the advance of deep learning and generative models, many techniques have been developed for image editing including the style transfer [2, 4, 5], the drawing based editing [6, 7], and the domain/attribute translation [8–10]. While manipulating images using these methods become easier as most operations are done automatically, editing images specifically per user's intention becomes more difficult.

The goal of this paper is to manipulate images using natural language description. We specifically focus on modifying visual attributes of an object, where the visual attributes are characterized by the color and the texture of an object. Fig. 1 illustrates our problem. In the examples, the colors of different parts of an bird are changed according to the given text description. We believe that it is a more intuitive way to manipulate images and would also be ideal for mobile devices.

Conditional deep generative models have shown great potential in multi-modal generative modeling of image and text. However, most existing studies concentrate on the text-to-image synthesis [11–14], which generates images from text descriptions without the original image. Only few works

This particular bird with a **red head and breast** and features **grey wings**.

This small bird has a **blue crown** and **white belly**.

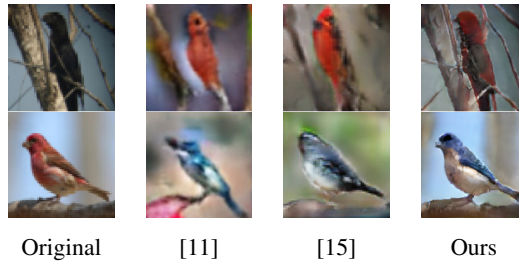

Original    [11]    [15]    Ours

Figure 1: Examples of image manipulation using natural language description. Existing methods produce reasonable results, but fail to preserve text-irrelevant contents such as the background of the original image. In comparison, our method accurately manipulates images according to the text while preserving text-irrelevant contents.

address similar problem as ours [11, 15]. In [11, 15], both approaches train generative adversarial networks (GANs) using the encoded image and the sentence vector pretrained for visual-semantic similarity [16, 17]. As shown in Fig. 1, these methods synthesize a new image according to the text while preserving the image layout and the pose of the object to some extent. However, they are likely to generate a new image conditioned on the pose of the original image instead of modifying only the parts that are described in the text. This is mainly because the sentence-conditional discriminator provides the generator with coarse training feedback, which prevents the generator from disentangling different regions of the image.

To overcome this limitation, we propose a novel text-conditioned visual attribute manipulation method called the Text-Adaptive Generative Adversarial Network (TAGAN). The key idea is to split a single sentence-level discriminator into a number of word-level discriminators so that each word-level discriminator is attached to a specific type of visual attribute. With this approach, the discriminator is able to provide fine-grained training feedback to the generator to change only the specific visual attribute. To this end, we introduce a text-adaptive discriminator that consists of word-level local discriminators. It classifies an image as real or fake by aggregating all word-level matching scores from the local discriminators with text attention. With the text-adaptive discriminator, our generator learns to change the parts of the image while preserving other contents. To the best of our knowledge, none of the previous works learn this level of fine-grained discriminators using text.

Experimental results show that our method outperforms existing methods on CUB [18] and Oxford-102 [19] datasets both quantitatively and qualitatively. We conducted a user study for human evaluation, and our results were preferred the most by the participants. In addition, our extensive analysis shows that the TAGAN effectively disentangles visual attributes and accurately manipulates images according to the text, while preserving text-irrelevant contents in the original image.

## 2   Related Work

With the success of deep generative models like variational auto-encoders (VAEs) [20] and GANs [21], image generation has been widely studied, and our work is particularly related to conditional image generation methods [22, 23].

There have been many attempts to generate images using conditional variables. cGAN [23] generates MNIST digit images from labels. Attribute2Image [24] produces more complex images such as faces and birds from visual attributes by disentangling the foreground and the background using cVAE. InfoGAN [25] learns interpretable latent variables in an unsupervised manner for conditional image generation. However, these works address the problem of generating new images from pure noise vectors. In comparison, our work focuses on modifying the given image according to the conditional information.

Another line of research is the conditional image manipulation. Zhu *et al*. [6] introduced modifying the color and the shape of images from user's sketch by manipulating latent vectors. Similarly, Brock *et al*. [7] also proposed to edit face images with user scribbles. FaderNetworks [8] learns the attribute-invariant latent space by preventing correct prediction of attributes for face manipulation.

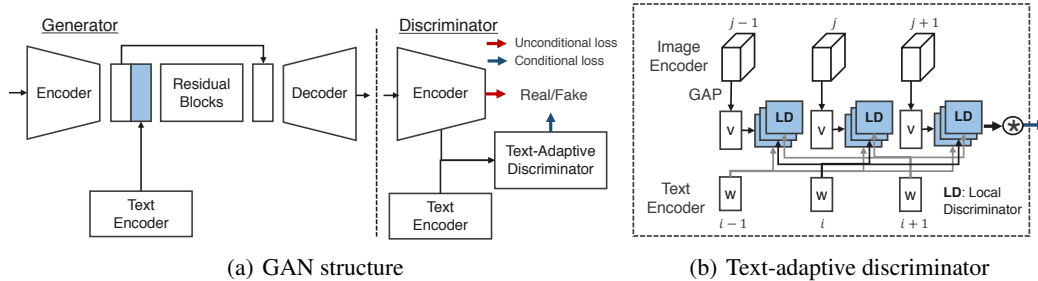

|  |  |
|---|---|
| (a) GAN structure | (b) Text-adaptive discriminator |

Figure 2: The proposed GAN structure. (a) shows the overall GAN architecture and (b) depicts our text-adaptive discriminator. In (b), the attention and the layer-wise weight are omitted for simplicity.

Additionally, image-to-image domain translation methods have been introduced [9, 10]. However, these methods cannot be directly applied to our task since they do not have mechanisms to process the sequential textual data as inputs. In this paper, we focus on the multi-modal learning of both image and natural language description.

Our work is also closely related to text-to-image synthesis methods. Reed *et al*. [11] first proposed to generate a 64×64 natural image from a sentence vector using the DCGAN. StackGAN [12, 26] produces high-resolution images progressively by stacking multiple GANs. Zhang *et al*. [14] presented a hierarchically nested adversarial loss to regularize the generator to improve image details. AttnGAN [13] incorporated an attention mechanism into [26] and enhanced fine-grained details with a pretrained visual-semantic similarity model. While it uses a word-level visual-semantic attention similar to our method, it fundamentally relies on a sentence vector to generate images. All of the methods produce various high-quality images from text, but their main focus is in generating a whole new image, not on manipulating a specific part of input image using text.

In [11], the authors briefly showed an image manipulation method as an application by learning an auxiliary image encoder to reconstruct a latent noise vector that contains text-irrelevant information of image. Dong *et al*.'s work [15] also addressed the problem of manipulating images with semantically different text descriptions. Their method trains a conditional GAN to synthesize a manipulated version of image given an original image and a target text description. However, both methods are limited in performance due to the simple sentence modeling discussed previously. In this paper, we tackle the problem in a different manner by making a fine-grained discriminator to learn to disentangle visual attributes from the image and the text.

## 3 Text-Adaptive Generative Adversarial Networks

Let $\mathbf{x}, \mathbf{t}, \hat{\mathbf{t}}$ denote an image, a positive text where the description matches the image, and a negative text that does not correctly describe the image, respectively. Given an image $\mathbf{x}$ and a target negative text $\hat{\mathbf{t}}$, our task is to semantically manipulate $\mathbf{x}$ according to $\hat{\mathbf{t}}$ so that the visual attributes of the manipulated image $\hat{\mathbf{y}}$ match the description of $\hat{\mathbf{t}}$ while preserving other information. We use GAN as our framework, in which the generator is trained to produce $\hat{\mathbf{y}} = G(\mathbf{x}, \hat{\mathbf{t}})$. Similar to text-to-image GANs [11, 15], we train our GAN to generate a realistic image that matches the conditional text semantically. In the following, we describe the TAGAN in detail.

**Generator**    The generator is an encoder-decoder network as shown in Fig. 2 (a)[1]. It first encodes an input image to a feature representation, then transforms it to a semantically manipulated representation according to the features of the given conditional text. For the text representation, we use a bidirectional RNN to encode the whole text. Unlike existing works [11, 15], we train the RNN from scratch, without pretraining. Additionally, we adopt the conditioning augmentation method [12] for smooth text representation and the diversity of generated outputs. As shown in Fig. 2 (a), manipulated contents are generated through several residual blocks with a skip connection. However, this process

may generate a new background and other contents that are not described in the text. Therefore, we use the reconstruction loss [27] when a positive text is given, which enforces the generator to reconstruct the text-irrelevant contents from the input image instead of generating new contents:

$$L_{rec} = \|\mathbf{x} - G(\mathbf{x}, \mathbf{t})\|. \tag{1}$$

However, learning invariant representation is still difficult unless the discriminator provides useful feedback for disentangling visual attributes. To cope with it, we propose a text-adaptive discriminator.

**Text-adaptive discriminator**    The motivation of the text-adaptive discriminator is to provide the generator with a specified training signal to generate certain visual attributes. To achieve this, the discriminator classifies each attribute independently using word-level local discriminators. By doing so, the generator receives feedback from each local discriminator for each visual attribute.

Fig. 2 (b) shows the structure of the text-adaptive discriminator. Similar to the generator, the discriminator is trained with its own text encoder. For each word vector $\mathbf{w}_i$, $i$-th output from the text encoder, we create 1D sigmoid local discriminator $f_{\mathbf{w}_i}$, which determines whether a visual attribute related to $\mathbf{w}_i$ exists in the image. Formally, $f_{\mathbf{w}_i}$ is described as:

$$f_{\mathbf{w}_i}(\mathbf{v}) = \sigma(\mathbf{W}(\mathbf{w}_i) \cdot \mathbf{v} + \mathbf{b}(\mathbf{w}_i)), \tag{2}$$

where $\mathbf{W}(\mathbf{w}_i)$ and $\mathbf{b}(\mathbf{w}_i)$ are the weight and the bias dependent on $\mathbf{w}_i$. $\mathbf{v}$ is an 1D image vector computed by applying global average pooling to the feature map of the image encoder.

With the local discriminators, the final classification decision is made by adding word-level attentions to reduce the impact of less important words to the final score. Our attention is a softmax values across $T$ words, which is computed by:

$$\alpha_i = \frac{\exp(\mathbf{u}^T \mathbf{w}_i)}{\sum_i \exp(\mathbf{u}^T \mathbf{w}_i)}, \tag{3}$$

where $\mathbf{u}$ is a temporal average of $\mathbf{w}_i$. The final score is computed according to the following formulation:

$$D(\mathbf{x}, \mathbf{t}) = \prod_{i=1}^{T} [f_{\mathbf{w}_i}(\mathbf{v})]^{\alpha_i}. \tag{4}$$

Our approach has several advantages over existing methods. First, instead of learning sentence-level correspondence, we train the discriminator to first identify individual attributes in a sentence, then to find existence of each attribute in the image. Also, our method can be easily trained by the cross-entropy loss without the ranking loss [11] due to our multiplicative aggregation of scores. Although our method shares similar spirit to the attention-driven matching score in [13], our method does not use explicit spatial attention on images and does not need any hand-tuned hyperparameters. Also, the most important difference is that we do not use our text-adaptive discriminator as an auxiliary loss. Instead, we train the text-adaptive discriminator as our core GAN framework.

We additionally consider multi-scale image features to make some attribute detectors to focus on small-scale features and others to focus on large-scale features. Therefore, our conditional discriminator is rewritten as:

$$D(\mathbf{x}, \mathbf{t}) = \prod_{i=1}^{T} [\sum_j \beta_{ij} f_{\mathbf{w}_i, j}(\mathbf{v}_j)]^{\alpha_i}, \tag{5}$$

where $\mathbf{v}_j$ is the image vector of $j$-th layer, and $\beta_{ij}$ is a softmax weight that determines the importance of the layer $j$ for each word $\mathbf{w}_i$.

**GAN objective**    The final GAN objective consists of unconditional adversarial losses for $D(\mathbf{x})$, text-conditional losses for $D(\mathbf{x}, \hat{\mathbf{t}})$, and a reconstruction loss as shown in Fig. 2. The discriminator has one image encoder and two branches of classifier on the top of the encoder to compute both the unconditional and the conditional losses. Our network is trained by alternatively minimizing both the discriminator and the generator objectives described as:

$$
\begin{aligned}
L_D = \;& \mathbb{E}_{\mathbf{x}, \mathbf{t}, \hat{\mathbf{t}} \sim p_{data}}[\log D(\mathbf{x}) + \lambda_1 (\log D(\mathbf{x}, \mathbf{t}) + \log(1 - D(\mathbf{x}, \hat{\mathbf{t}})))] \\
& + \mathbb{E}_{\mathbf{x}, \hat{\mathbf{t}} \sim p_{data}}[\log(1 - D(G(\mathbf{x}, \hat{\mathbf{t}})))],
\end{aligned}
\tag{6}
$$

$$L_G = \mathbb{E}_{\mathbf{x},\hat{\mathbf{t}} \sim p_{data}}[\log D(\mathbf{x}) + \lambda_1 \log D(G(\mathbf{x},\hat{\mathbf{t}}),\hat{\mathbf{t}})] + \lambda_2 L_{rec}, \qquad (7)$$

where $\lambda_1$ and $\lambda_2$ control the importance of additional losses, and $\hat{\mathbf{t}}$ is randomly sampled from a dataset regardless of $\mathbf{x}$. Note that we do not penalize generated outputs using the conditional discriminator in Eq. (6) due to instability of training. In our experiment, our objective was enough to produce real images having manipulated attributes.

**Implementation** We implemented our method using PyTorch. For the generator, we adopt the architecture of the generator from [15] to encode and decode 128×128 images. We additionally encode a text using a bidirectional GRU and the pretrained fastText [28] word vectors. In the discriminator, we use `conv3`, `conv4`, and `conv5` for the local discriminators.[2] We trained our network 600 epochs using Adam optimizer [29] with the learning rate of 0.0002, the momentum of 0.5, and the batch size of 64. Also, we decreased the learning rate by 0.5 for every 100 epochs. For data augmentation, we used random cropping, flipping, and rotation. We resized images to 136×136 and randomly cropped 128×128 patches. The random rotation ranged from -10 to 10 degrees. We set $\lambda_1$ and $\lambda_2$ to 10 and 2 respectively considering both the visual quality and the training stability.

## 4 Experiments

**Experimental setup** We evaluated our method on CUB dataset [18] and Oxford-102 dataset [19], which contain 11,788 bird images of 200 categories and 8,189 flower images of 102 categories, respectively. Also, we used natural language captions of both datasets provided in [11], where 10 individual sentences were annotated for each image. For evaluation, we compared our method to two baseline methods: SISGAN [15] and AttnGAN [13]. In particular, we adapted AttnGAN for our task based on the source code provided by the authors in order to compare our method with the state-of-the-art text-to-image synthesis method. Specifically, we added an image encoder and a reconstruction loss to the original network. For the encoder, we used the same encoder used in our method. We fine-tuned the weight of the reconstruction loss to produce the best results. For SISGAN, we reproduced the same network based on the original paper and the code provided by the authors.

**Quantitative results** We first conducted a quantitative evaluation of the two baseline methods and our method. The quality of synthesized images is a subjective matter, and it is difficult to compare different methods using an objective metric. The inception score [30] is also not proper for this problem since it only measures the quality of an image, not the relevance to a text. Therefore, we conducted a human evaluation on Amazon Mechanical Turk.

For the user study, we randomly selected 10 images and 10 texts from the test set, and produced 200 outputs from the two datasets for each method. As the spatial resolutions for different methods are different, we resized all output images to 64×64 to prevent the users from evaluating the images based on the sharpness. Then, we asked workers to rank three results after looking at both input image/text and outputs. We asked the workers to evaluate images based on the following two criteria: (i) whether the visual attributes (colors, textures) of the manipulated image match the text, and the background irrelevant to the text is preserved, and (ii) whether the manipulated image looks natural, and visually pleasing. As a result, we collected a total of 4,000 samples from 20 workers.

Table 1 shows the results of the user study. In the table, each criterion is referred as (i) Accuracy and (ii) Naturalness. The values shown in the table are average ranking values. For both the accuracy and the naturalness, our results were most preferred by the workers. To further verify it, we conducted chi-square test on the count values and found that p-value is $p < 10^{-30}$, which indicates that our method significantly outperforms baseline methods on this user study. In essence, our method generates realistic images, where the visual attributes are manipulated accurately while preserving text-irrelevant contents of the original image. The performance of two baseline methods were similar, but AttnGAN was slightly more preferred over SISGAN.

To further compare the quality of the content preservation, we also computed $L_2$ reconstruction error by forwarding images with positive texts. Due to the randomness, we average 50 trials for each method. In the table, our method shows the lowest reconstruction error, which indicates that our method preserves the content of original image better.

Table 1: Quantitative comparison. Accuracy and Naturalness were evaluated by users, and the values indicate the average ranking. $L_2$ reconstruction error was additionally compared.

| Method | CUB | | | Oxford-102 | | |
|---|---|---|---|---|---|---|
| | Accuracy | Naturalness | $L_2$ error | Accuracy | Naturalness | $L_2$ error |
| SISGAN [15] | 2.33 | 2.34 | 0.30 | 2.67 | 2.28 | 0.29 |
| AttnGAN [13] | 2.19 | 2.11 | 0.25 | 2.21 | 2.10 | 0.32 |
| Ours | **1.49** | **1.56** | **0.11** | **1.52** | **1.62** | **0.11** |

Original

This bird has **wings that are blue** and has a **white belly**.

A small bird with **white base** and **black stripes** throughout its belly, head, and feathers.

Original

The petals of the flower have **yellow and red stripes**.

This flower has petals of **pink and white color** with **yellow stamens**.

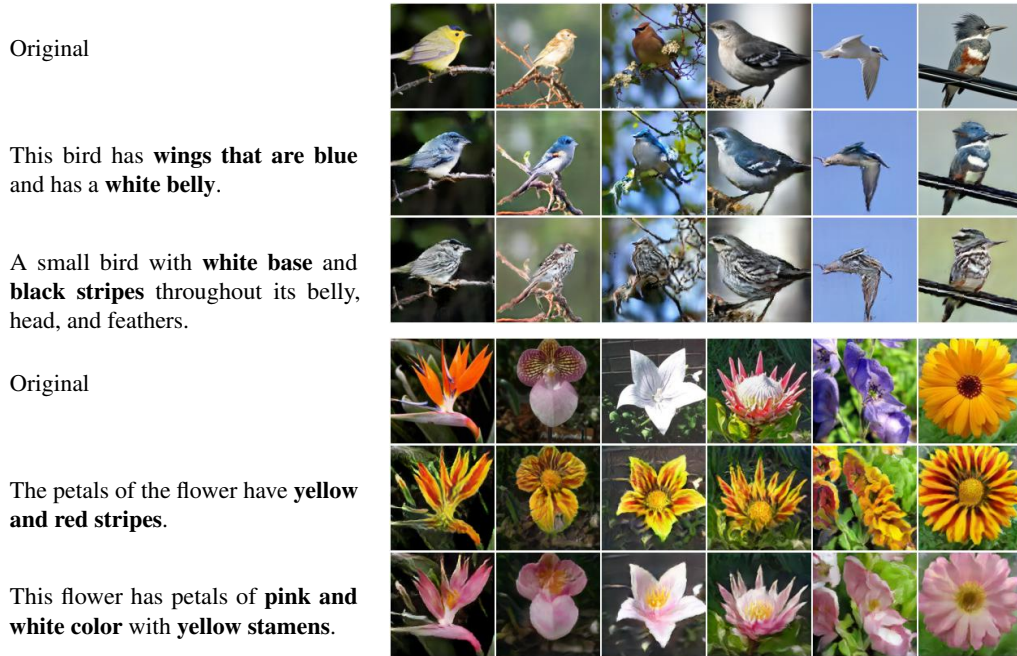

Figure 3: Qualitative results of our method on CUB and Oxford-102 datasets.

This is a **black bird** with **gray and white wings** and a **bright yellow belly and chest**.

This flower has **petals that are white** and has **patches of yellow**.

Original

SISGAN [15]

AttnGAN [13]

Ours

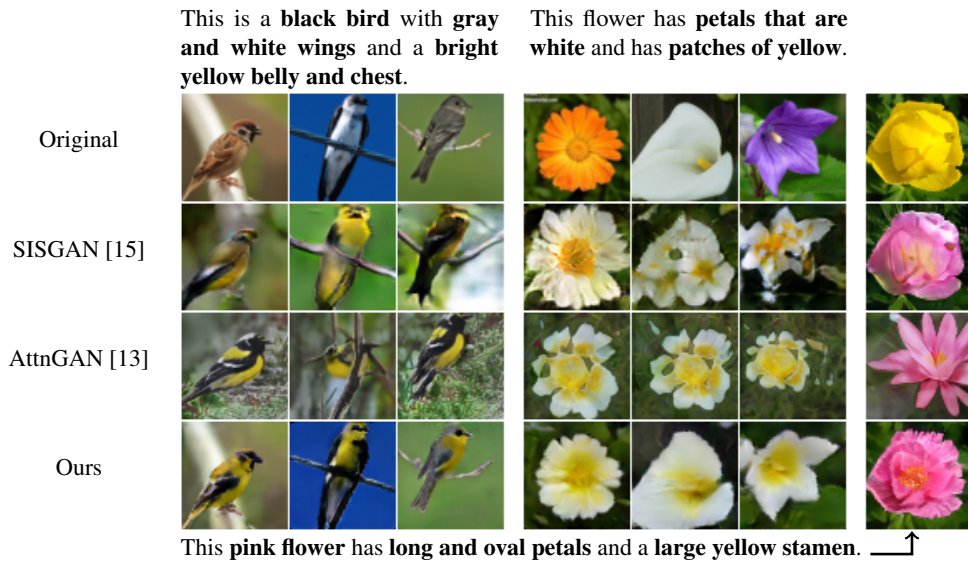

This **pink flower** has **long and oval petals** and a **large yellow stamen**.

Figure 4: Qualitative comparison of three methods. In most cases, our method outperforms baseline methods qualitatively. The rightmost column shows a failure case using our method.

This bird is brown with black wings and tail and long legs.

This flower has petals that are yellow and are very stringy.

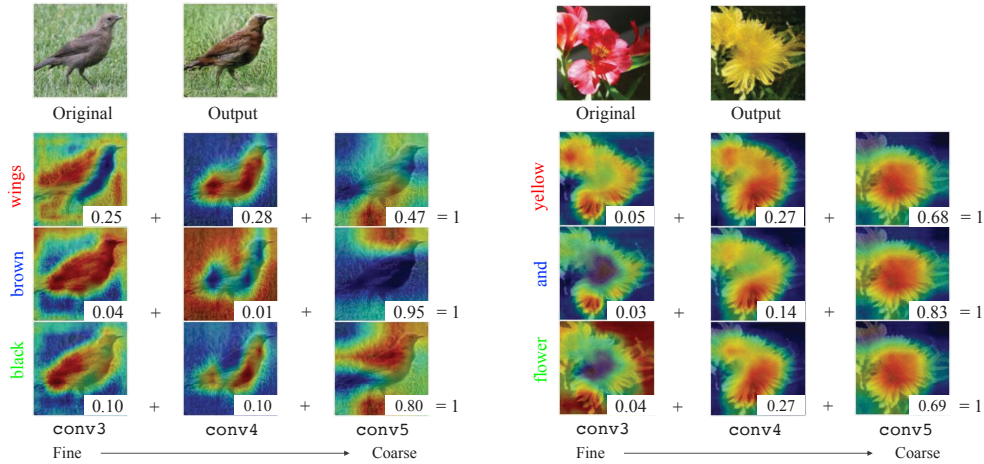

Figure 5: Visualization of the text-adaptive discriminator. From top to bottom, the top-3 word attentions are shown. From left to right, the saliency maps of 3 layer-wise local discriminators are visualized. Each fractional number is $\beta_{ij}$. Note that $\sum_j \beta_{ij} = 1$.

**Qualitative results**   Fig. 3 shows qualitative results of our method on CUB and Oxford-102. In most cases, the visual attributes of images are accurately changed according to the text. Note that some attributes only exist in a specific class of birds or flowers. For example, the petals of "yellow and red stripes" in Fig. 3 refers to the flower *gazania*. As can be seen from the results, our method is able to generate textures on images of different classes. It indicates that our network effectively disentangles visual attributes invariant to pose, shape, and background.

Fig. 4 shows comparisons with the baselines methods. Both baselines usually preserve the pose and the layout of the original images. However, the methods are likely to generate a new image based on the original layout and do not preserve the contents that are not relevant to the text. In some cases, the methods generate similar images because the sentence is highly correlated to a particular class of birds or flowers. On the other hand, our method preserves the text-irrelevant contents while transferring visual attributes accurately.

Our method may sometimes generate failure cases. As shown in the rightmost column in Fig. 4, our method fails to generate the exact shape described in the text on the original image. It is mainly due to the trade-off between generating new contents and preserving original contents. While we can control it by adjusting the power of reconstruction loss, some examples need more sophisticated control. Similar to [27, 8], our potential extension is to incorporate an explicit control variable into our generator to decide the trade-off example by example.

**Component analysis**   As described in Sec. 3, our text-adaptive discriminator learns both word-level attention and local discriminators. In Fig. 5, we visualized class activation maps (CAMs) [31] to show the saliency map that a local discriminator in Eq. (2) attends to. [3] The figure shows the top-3 word attentions vertically and 3 layer-wise weights horizontally. We can see that the discriminator learns distinguishable visual attributes in the text and their corresponding classifiers. Additionally, the network adaptively selects different levels of layers to detect coarse or fine-grained attributes for each word. To show the effectiveness of it, we also ran an ablation study as shown in Fig. 6. When we use only `conv5`, the network usually learns the global object colors. The network generates more details when trained using `conv3`, but the quality is not satisfying as various scales of visual attributes are hard to learn within a single scale layer. On the other hand, our multi-scale network (`conv3,4,5`) effectively learns to generate both coarse and fine-grained visual attributes by separating different scales of the attributes in the latent space.

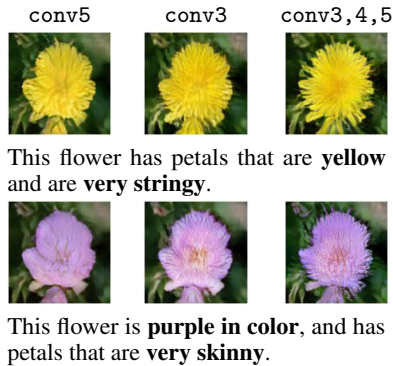

This flower has petals that are **yellow** and are **very stringy**.

This flower is **purple in color**, and has petals that are **very skinny**.

Figure 6: Ablation study of multi-scale layers.

Table 2: Quantitative comparison of multi-modal retrieval task on CUB dataset. Our method is competitive to the state-of-the-art method [32]

|  | Image-Text | Text-Image |
|---|---|---|
|  | Top-1 Acc (%) | AP@50 (%) |
| [34] | 56.8 | 48.7 |
| [32] | **61.5** | 57.6 |
| [13] | 55.1 | 51.0 |
| Ours | 61.3 | **62.8** |

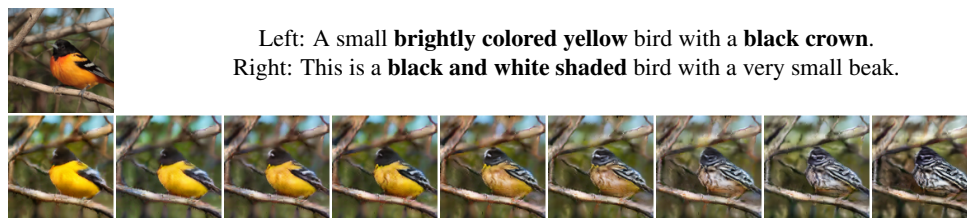

Left: A small **brightly colored yellow** bird with a **black crown**.
Right: This is a **black and white shaded** bird with a very small beak.

Figure 7: Sentence interpolation results. Our generator smoothly generates new visual attributes without loosing original image.

Since our method shares similar word attention paradigm with image-text matching methods [32, 13], we further provide a comparison on this task. We trained our text-adaptive classifier on top of the Inception-v3 network [33] to compute the image-text similarity scores. Similar to [34], we compared the top-1 image-to-text retrieval accuracy (Top-1 Acc), and the percentage of the matching images in the top-50 text-to-image retrieval results (AP@50) on CUB dataset as shown in Table 2. Our method outperforms the auxiliary matching network in AttnGAN [13], and is competitive with the state-of-the-art matching method [32]. We attribute this result to the word-level local discriminators, and the use of multi-scale layers. In addition to performance, our method is more convenient to train since it does not need hard sample mining [32], and data-dependent hyperparameters [13].

Lastly, we conducted a text interpolation experiment for the generator to analyze the inference ability. We interpolated a text embedding vector in the generator using two different sentences, and generated smoothly changing images. As shown in Fig. 7, our method generates reasonable images of interpolated text while preserving the contents of original image. This indicates that our generator effectively learns invariant latent variables and is able to infer with respect to various texts without memorizing them.

## 5    Conclusion

In this paper, we proposed a text-adaptive generative adversarial network to semantically manipulate images using natural language description. Our text-adaptive discriminator disentangles fine-grained visual attributes in the text using word-level local discriminators created on the fly according to the text. By doing so, our generator learns to generate particular visual attributes while preserving irrelevant contents in the original image. Experimental results show that our method outperforms existing methods both quantitatively and qualitatively.

**Acknowledgement**   This work was supported by Global Ph.D. Fellowship Program through the National Research Foundation of Korea (NRF) funded by the Ministry of Education (NRF2015H1A2A1033924), Institute for Information & communications Technology Promotion (IITP) grant funded by the Korea government (MSIP) (2018-0-01858, Video Manipulation and Language-based Image Editing Technique for Detecting Manipulated Image/Video), and the ICT R&D program of MSIT/IITP (2017-0-01772, Development of QA systems for Video Story Understanding to pass the Video Turing Test).

## Footnotes

[1]Note that generating high-resolution images is not the main focus of our paper, therefore we do not use multi-stage generation [12–14, 26] in our generator.

[2]For the detail of the network architecture, please refer to the supplementary material.

[3]Since the grid of `conv5` is coarse (4×4), the salient region may not exactly match to an object region.

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
