[Supplementary Material]

# (Supplementary Material)
# Text-Adaptive Generative Adversarial Networks: Manipulating Images with Natural Language

**Seonghyeon Nam, Yunji Kim, and Seon Joo Kim**
Yonsei University
{shnnam,kim_yunji,seonjookim}@yonsei.ac.kr

## 1  Additional Experimental Results

In this supplementary material, we show additional results of our method. Fig. 1 and Fig. 2 show qualitative comparison on CUB and Oxford-102 datasets, and Fig. 3 and Fig. 4 show additional qualitative results of our method.

## 2  Network Architecture

Table 1 and 2 show the hyperparameters of the proposed network. For the conditional discriminator, we added additional $3\times3$ convolution layers to `conv3`, `conv4`, and `conv5` layers in the unconditional discriminator. Then, the features from those layers are spatially reduced by global average pooling and classified by local discriminators. The parameters of each local discriminator is generated from each word vector of the RNN. (Conv2d(K, P): 2D convolution with the kernel size K and the padding P, BN: Batch normalization, LeakyReLU(S): LeakyReLU with the negative slope S, NN Upsampling: Nearest neighbor upsampling)

## References

[1] H. Zhang, T. Xu, H. Li, S. Zhang, X. Wang, X. Huang, and D. Metaxas, "Stackgan: Text to photo-realistic image synthesis with stacked generative adversarial networks," in *ICCV*, 2017.

[2] H. Dong, S. Yu, C. Wu, and Y. Guo, "Semantic image synthesis via adversarial learning," in *ICCV*, Oct 2017.

[3] Q. H. H. Z. Z. G. X. H. X. H. Tao Xu, Pengchuan Zhang, "Attngan: Fine-grained text to image generation with attentional generative adversarial networks," in *CVPR*, 2018.

Table 1: The parameters of the generator.

| Module | Layers | Input size | Output size |
|---|---|---|---|
| Text Encoder | Bidirectional GRU | # of words $\times$ 300 | # of words $\times$ 512 |
|  | Temporal Averaging | # of words $\times$ 512 | 512 |
|  | Linear, LeakyReLU(0.2) | 512 | 256 |
| (a) | Conditioning Augmentation [1] | 256 | 128 |
| Image Encoder | Conv2d(3, 1), ReLU | 3$\times$128$\times$128 | 64$\times$128$\times$128 |
|  | Conv2d(4, 2), BN, ReLU | 64$\times$128$\times$128 | 128$\times$64$\times$64 |
|  | Conv2d(4, 2), BN, ReLU | 128$\times$64$\times$64 | 256$\times$32$\times$32 |
| (b) | Conv2d(4, 2), BN, ReLU | 256$\times$32$\times$32 | 512$\times$16$\times$16 |
| Concat (a) and (b) | Conv2d(3, 1), BN, ReLU | 640$\times$16$\times$16 | 512$\times$16$\times$16 |
| Residual Blocks | 4 $\times$ Residual Block (below) | 512$\times$16$\times$16 | 512$\times$16$\times$16 |
| Residual Block | Conv2d(3, 1), BN, ReLU | 512$\times$16$\times$16 | 512$\times$16$\times$16 |
| (c) | Conv2d(3, 1), BN | 512$\times$16$\times$16 | 512$\times$16$\times$16 |
|  | Input + (c) | 512$\times$16$\times$16 | 512$\times$16$\times$16 |
| Decoder | NN Upsampling (2$\times$) | 512$\times$16$\times$16 | 512$\times$32$\times$32 |
|  | Conv2d(3, 1), BN, ReLU | 512$\times$32$\times$32 | 256$\times$32$\times$32 |
|  | NN Upsampling (2$\times$) | 256$\times$32$\times$32 | 256$\times$64$\times$64 |
|  | Conv2d(3, 1), BN, ReLU | 256$\times$64$\times$64 | 128$\times$64$\times$64 |
|  | NN Upsampling (2$\times$) | 128$\times$64$\times$64 | 128$\times$128$\times$128 |
|  | Conv2d(3, 1), BN, ReLU | 128$\times$128$\times$128 | 64$\times$128$\times$128 |
|  | Conv2d(3, 1), Tanh | 64$\times$128$\times$128 | 3$\times$128$\times$128 |

Table 2: The parameters of the discriminator.

| Module | Layers | Input size | Output size |
|---|---|---|---|
| Image Encoder | Conv2d(4, 2), LeakyReLU(0.2) | 3$\times$128$\times$128 | 64$\times$64$\times$64 |
|  | Conv2d(4, 2), BN, LeakyReLU(0.2) | 64$\times$64$\times$64 | 128$\times$32$\times$32 |
| conv3 | Conv2d(4, 2), BN, LeakyReLU(0.2) | 128$\times$32$\times$32 | 256$\times$16$\times$16 |
| conv4 | Conv2d(4, 2), BN, LeakyReLU(0.2) | 256$\times$16$\times$16 | 512$\times$8$\times$8 |
| conv5 | Conv2d(4, 2), BN, LeakyReLU(0.2) | 512$\times$8$\times$8 | 512$\times$4$\times$4 |
| Unconditional Discriminator | Conv2d(4, 0), Softmax | 512$\times$4$\times$4 | 1$\times$1$\times$1 |
| Text Encoder | Bidirectional GRU | # of words $\times$ 300 | # of words $\times$ 512 |
| $\beta_{ij}$ | Linear, Softmax | # of words $\times$ 512 | # of words $\times$ 3 |
| $\alpha_i$ | See Eq. (3) in the paper | # of words $\times$ 512 | # of words $\times$ 1 |
| $f_{\mathbf{w}_i,j}$ | Linear (See Eq. (2) in the paper) | N/A | N/A |
| From conv3 | Conv2d(3, 1), BN, LeakyReLU(0.2) | 256$\times$16$\times$16 | 256$\times$16$\times$16 |
| (a) | Global Average Pooling | 256$\times$16$\times$16 | 256$\times$1$\times$1 |
| From conv4 | Conv2d(3, 1), BN, LeakyReLU(0.2) | 512$\times$8$\times$8 | 512$\times$8$\times$8 |
| (b) | Global Average Pooling | 512$\times$8$\times$8 | 512$\times$1$\times$1 |
| From conv5 | Conv2d(3, 1), BN, LeakyReLU(0.2) | 512$\times$4$\times$4 | 512$\times$4$\times$4 |
| (c) | Global Average Pooling | 512$\times$4$\times$4 | 512$\times$1$\times$1 |
| Conditional Discriminator | See Eq. (5) in the paper with ($\alpha_i$, $\beta_{ij}$, $f_{\mathbf{w}_i,j}$, (a), (b), (c)) | N/A | 1$\times$1$\times$1 |

This bird has **wings that are black and white** and has a **red tinted face**.

This is a **black bird** with **gray and white wings** and a **bright yellow belly and chest**.

This bird has **wings that are brown** and has a **white belly**.

Figure 1: Qualitative comparison on CUB dataset.

This flower has **orange petals** that has **yellow shading in the center**.

Original

SISGAN [2]

AttnGAN [3]

Ours

This flower has **petals that are white** and has **patches of yellow**.

Original

SISGAN [2]

AttnGAN [3]

Ours

This flower is **pink, white, and yellow in color**, and has **petals that are multi colored**.

Original

SISGAN [2]

AttnGAN [3]

Ours

Figure 2: Qualitative comparison on Oxford-102 dataset.

Original

This bird has **wings that are brown** and has a **white belly**.

The bird has **mostly blue plumage** with **streaks of dark grey on the wings and tail**.

This bird has **wings that are grey** and has a **white belly**.

This bird has **wings that are grey** and has a **yellow belly**.

Original

A small bird with **brown and black feathers**, **white belly**, **white eyering**, and a small **brown beak**.

A small **black and white** bird with a long tail, long black legs, a **white belly**, a small head, and a short pointy beak.

This bird is **red with blue** and has a long, pointy beak.

This is a small bird with a **white belly**, a **black and white spotted back** and a pointed beak.

This bird has a **black body** with an **orange beak**.

Figure 3: Additional qualitative results of our method on CUB dataset.

Original

This flower is **white and yellow** in color, with oval shaped petals.

This flower has **white petals** with **pink on the edges of them**.

This flower has a **wide yellow center** surrounded by **long yellow petals** with **central red stripes**.

This flower has **petals that are red** and has **yellow tips**.

This flower has **petals that are pink** and has **yellow stamen**.

Original

This flower has **white petals** with a **splash of red coloring** in the middle of each one.

The petals on this flower are **white** with **yellow stamen**.

This flower is **yellow and brown** in color, with petals that are oval shaped.

This flower has petals that are **pink** and has **yellow stamen**.

This flower has petals that are **white** and has **a peach style**.

Figure 4: Additional qualitative results of our method on Oxford-102 dataset.