[Reviews · NeurIPS 2018]

Reviewer 1



After rebuttal comments: * readability: I trust the authors to update the paper based on my suggestions (as they agreed to in their rebuttal). * reproducibility: the authors gave clear guidelines in the rebuttal of how their produced their results. For AttrGAN, they did change the weight sweep and for SISGAN they used the same hyperparameters as they used in their method (which I would object to in general, but given that the authors took most of their hyperparameters from DCGAN, does not create an unfair advantage). I expect the additional details of the experimental results to be added in the paper (as supplementary material). ################### Main idea: ------------------ Task: Manipulate images based on text instruction. Ensure that content that is not relevant to the text does not change. Method: to avoid changing too much of the image, use local discriminators that learn the presence of individual visual attributes. The discriminator is a weighted combination of the local discriminators which determine whether a visual attribute (feature map in a convolutional image encoder) is related to a particular word. The local discriminators are combined using attention on the word vectors. Results: substantially improving on state of the art on modifying images based on text descriptions. However, the details of how the baselines are trained are not specified (see below). Reviewer knowledge: I am familiar with the GAN literature, but prior to reading this paper I have not been aware of work in manipulating images with text. I was not familiar with baselines (SISGAN and AttnGAN) before reading this paper. Comment on my rating: I am willing to increase my review substantially if the paper clarity substantially increases (see below) and if more details about the baseline training are released. Quality: -------- The idea and the experiments presented in the paper are interesting, and definitely worth noting. However, the paper suffers due to a lack of clarity (see below). Paper not fully reproducible and it is not clear how the baselines were trained: details of data augmentation are not fully specified (lines 171-172), details of training are also lacking, especially for the baselines. Were the same architectures used, or were the ones in the original papers used? Where the hyperparameters tuned on the presented method (this would severely cause a disadvantage to the baselines)? Specifically for SISGAN, the SISGAN paper introduces two version (with and without VGG) and it is not clear to which this paper is comparing. Originality ----------- Comparison with prior work: Not sufficient in the paper, but by reading the two baseline papers, SISGAN and AttnGAN, I believe the current work introduces the following differences: 1) Difference compared to AttnGAN * AttnGAN does not work with text, but with attributes (the goal is to generate an image with a an attribute change from the original image). This is a simpler task, so there is no need for local discriminators. * AttnGAN also uses a reconstruction loss, and an unconditional adversarial loss (using WGAN). The difference lies in the conditional version of the model, where in the presented work a conditional GAN is used, while AttnGAN uses an additional classifier which ensures that the attributes are preserved. 2) Difference compared to SISGAN SISGAN tackles the same problem as the one in this paper: changing images based on text instructions. The approach of combining the image and text embeddings in the encoder through residual blocks is used in both papers. in SISGAN, there is no unconditional discriminator - the discriminator sees pairs of fake data with real text as “fake”. The advantage of the current approach compared to SISGAN is that it ensures that the generated images are realistic, independent on the text association. SISGAN does not use a local discriminator that looks at individual words. The advantage of the presented method is the attention and the focus on individual words, which helps with generalization to new sentences. The paper does introduce a novel idea, of the text adaptive discriminator. The discriminator learns association between individual words and individual feature maps of the image feature learner, and combines these into creating a global discriminator. It is not clear to me why in equation (3), the attention is taken over the average temporal word embedding. Would using a learned attention structure here not be better? Significance: ------------- The results in the paper are impressive. Compared to the baselines, more content is preserved, and only text specific information is changed. The images are impressive, and the results in Table 1 show an improvement in quality based on human evaluation on a large number of samples. The paper also shows that the results are applicable for image-text similarity, as seen in Table 2. They improve or compete with state of the art in that domain as well. One important point of the paper is that it performs analysis on the presented method. First, they look at at the text adaptive discriminator, and visualizing the class activation maps per layer. By using class activation maps, one can see to see how the model is focusing on different aspects of the image at different locations in the network. Second, they perform sentence interpolation on the text to see the different generated images, and show interpretable results. The results here are impressive too, with the model showing generalization in embedding space. Though this is limited to the particular image shown in Figure 7, it is an interesting result. Clarity: ------ Overall: This is where the paper lacks most. The core ideas need more detail, and core concepts are not explained. Concrete example: “visual attributes" are concretely defined late in the paper, and there vaguely too. The set up of the paper is not clearly described early in the paper (what is a “positive” and what is a “negative” example text?), and the reader has to infer this from the content later on. The datasets used have (x, t) examples of images and text relevant to that image. These are the positive examples. The negative examples are obtained by sampling another t from the data. The most relevant related work is also vaguely described (I had to read the two prior relevant papers to understand the differences). Detailed feedback: The introduction section which describes the proposed method (46-58) could have been sharper. I felt there was either not enough detail, or too much detail, depending on the view the authors wanted to take. Perhaps a rewrite would just fix this impression. Line 94: typo: to reconstruct a latent noise vector. Discussion of related work: This discussion could have been made clear by expanding on the related work in 93-100. This is by far the most relevant prior work, but gets less attention than the other prior work described. Lines 111-112: “we train the RNN in the generator from scratch. “ Perhaps “we train the RNN from scratch, without pretraining.” Lines 115-116: “Therefore, we use the reconstruction loss [27] when positive pairs of the image and the text are forwarded”. So far it is not clear what the positive examples are, so this sentence is confusing. If the problem statement was more clearly defined, this would not be an issue. One needs to state that positive examples are x and t that refer to the same content, taken from the dataset. Lines 120-127: The definition of “visual attribute" should be made clearer here. $v$ is used to represent the visual attribute, so perhaps its definition should be moved at the end of line 127. Line 179: “Although AttnGAN was originally designed for the text-to-image synthesis”. I think there is a not missing there. Lines 183-185 “The inception score [30] is also not proper for this problem since the mix-up of inter-class attributes may lower the score.” This sentence is not clear - but I agree that in this case Inception score will only try to measure image quality, not relevance to text. Lines 228-229: “the saliency map that a world classifier attends to”. Again, here it could be made more clear that this refers to the component classifiers of the discriminator, that learn to classify per word embedding and visual map (Equation 3). When describing figure 5, it would be useful to point out that the floating point numbers are the betas referred to in equation 5.

Reviewer 2



This is a very natural, but still technically involved extension of the conditional GAN principle the viability of which is supported by experimental evidence in the paper. The author(s) suggest a system that is able to transform images based on textual inputs. One interesting, implicit aspect of this work is that it demonstrates language grounding in actual actions in a procedural manner. The main idea of the paper is based on an text-adaptive discriminator utilizing a simple but efficient attention mechanism. This allows for identifying various attributes of the sentence and generalize from simpler sentences to more complicated ones. This idea allows for utilizing unconditional GAN losses for the discriminator and generator. The results are evaluated quantitatively by human raters using mechanical turk. This is a very impressive work with strong underlying ideas and impressive results and can be viewed as an important milestone in the application of generative architectures.

Reviewer 3



This paper tackles the problem of manipulating images by sentences description. To preserve the text-irrelevant content in the original image, they propose text-adaptive discriminator that creates word level local discriminators. The experimental results show that the proposed method can better manipulate the attributes while preserving text-irrelevant content in an image than other state-of-the-art text-to-image synthesis approaches. This paper is very well-written. It proposes a new task and a novel model to solve it. Both quantitative and qualitative results are convincing. Its contribution is significant and I recommend to accept it.

Reviewer 4



Considering rebuttal comments: Regarding the scalability of the proposed method, I understand that the network is enforced to consider phrase-level expression from jointly computing the final discriminator score from independent local discriminators. Also, the authors clearly explained how there is little computational complexity of long text input compared to number of pixels. Both concerns should be clearly explained in the final version of the paper. ---------------------------------------------- This paper deals with the task of semantically modifying visual attributes of an image using text description. Previous studies have dealt with synthesizing realistic images with text, but modifying specific attributes of an image while maintaining the image content is fairly a novel approach. Although there have been previous studies that manipulate images with text, this work is able to maintain text-irrelevant contents of the original image through a text-adaptive discriminator that adopts local discriminators for each word vector. As a result, this work can only modify specific attributes of an image through text without creating an entirely new (irrelevant) image. However, this paper seems to have limitations in terms of scalability. First all, the local discriminators learn each visual attribute related to each word vector “independently”. Due to this structure, TAGAN cannot understand more complex, phrase-level expressions that must consider semantics of “combination of words”. Thus, treating each word independently and only looking at fine-grained word-level semantics may rather deter the model’s understanding of semantic compositionality and produce incorrect manipulations. Furthermore, computational complexity of the model would scale by number of words in a sentence due to its local discriminator architecture. This will become highly inefficient as the length of the sentence increases. The proposed method may have worked for simple, clear sentences such as CUB dataset and Oxford-102 dataset, but it is doubtful that it will scale to more complex, longer text data. In this respect, I think the paper is marginally below the acceptance threshold.